# Coordination Polymers Based on a Biphenyl Tetraphosphonate Linker: Synthesis Control and Photoluminescence

**DOI:** 10.3390/molecules25081835

**Published:** 2020-04-16

**Authors:** Ana D. G. Firmino, Ricardo F. Mendes, Duarte Ananias, Jéssica S. Barbosa, João P. C. Tomé, Filipe A. Almeida Paz

**Affiliations:** 1Department of Chemistry, CICECO—Aveiro Institute of Materials, University of Aveiro, 3810-193 Aveiro, Portugal; danielafirmino1@ua.pt (A.D.G.F.); rfmendes@ua.pt (R.F.M.); dananias@ua.pt (D.A.); jessicambarbosa@ua.pt (J.S.B.); 2QOPNA and LAQV-REQUIMTE, Department of Chemistry, University of Aveiro, 3810-193 Aveiro, Portugal; jtome@tecnico.ulisboa.pt; 3Department of Physics, CICECO—Aveiro Institute of Materials, University of Aveiro, 3810-193 Aveiro, Portugal; 4CQE and Departamento de Engenharia Química Instituto Superior Técnico, Universidade de Lisboa Av. Rovisco Pais, no 1, 1049-001 Lisboa, Portugal

**Keywords:** metal-organic framework, crystal structure, photoluminescence

## Abstract

In this work, we used the rigid tetrapodal organic linker, [1,1′-biphenyl]-3,3′,5,5′-tetrayltetrakis(phosphonic acid) (H_8_btp), for the preparation of two lanthanide–organic framework families of compounds: layered [Ln_7_(H_5_btp)_4_(H_5.5_btp)_2_(H_6_btp)_2_(H_2_O)_12_]∙23.5H_2_O∙MeOH [where Ln^3+^ = Eu^3+^ (**1Eu**) and Gd^3+^ (**1Gd**)], prepared using microwave-irradiation followed by slow evaporation; 3D [Ln_4_(H_3_btp)(H_4_btp)(H_5_btp)(H_2_O)_8_]∙3H_2_O [where Ln^3+^ = Ce^3+^ (**2Ce**), Pr^3+^ (**2Pr**), and Nd^3+^ (**2Nd**)], obtained from conventional hydro(solvo)thermal synthesis. It is shown that in this system, by carefully selecting the synthetic method and the metal centers, one can increase the dimensionality of the materials, also increasing structural robustness (particularly to the release of the various solvent molecules). Compound **1** is composed of 2D layers stacked on top of each other and maintained by weak π–π interactions, with each layer formed by discrete 1D organic cylinders stacked in a typical brick-wall-like fashion, with water molecules occupying the free space in-between cylinders. Compound **2,** on the other hand, is a 3D structure with small channels filled with crystallization water molecules. A full solid-state characterization of **1** and **2** is presented (FT-IR spectroscopy, SEM microscopy, thermogravimetric studies, powder X-ray diffraction and thermodiffractometry). The photoluminescence of **1Eu** was investigated.

## 1. Introduction

Metal-Organic Frameworks (MOFs), or Porous Coordination Polymers (PCPs), are a class of crystalline materials built from the coordination of organic linkers to metal ions or clusters [1]. Among the assorted reasons why there are so many reports in the area of MOFs and PCPs, and why they remain an attractive topic of research, one can point out their tunable chemical structure and/or their multifunctional properties [2]. The ability to design a network with specific pore sizes and shapes is one of the main features why these materials are good candidates for multiple applications in different research areas like gas storage and separation [3,4], catalysis [5,6], sensing [7,8,9,10,11], and drug delivery [12,13], among others [14,15,16,17,18,19,20,21].

While the main research regarding MOFs remains focused on materials derived from *d*-block metals coordinated to carboxylate-based organic linkers, our research group has focused on metal phosphonates, more specifically, lanthanide phosphonate-based MOFs (LnOFs). LnOFs have gained particular interest, in some part due to the possibility to design new materials by changing the lanthanide radii. In some particular cases, the LnOFs properties can also change [22,23,24]. Recently, Wang and co-workers were able to improve the DNA sensing of LnOF nanosheets by simple change in lanthanide ions [25]. The contraction of ionic radii led to a lower quenching efficiency. Tetraphosphonic organic linkers have gained particular interest in recent years, especially phosphonic acids based on tetraphenylmethane and tetraphyenylsilane building units. [26] While their geometries can pose some challenges regarding its synthetic approaches (mainly due to the high rigidity that these molecules present), it results in their properties improvement. A particular improvement is the stability of these materials: while many MOFs show low stability in even ambient conditions, some of these materials can withstand solutions of concentrated acids (HCl and H_2_SO_4_) and even *aqua regia* [27]. In our group, the rich chemistry of phosphonic acid allowed the preparation of a variety of structures ranging from 1D to 3D networks by, in many cases, slight changes in experimental conditions. With this we were able to synthetize MOFs with a broad range of applications: from photoluminescence [28,29,30], to luminescent thermometers [31], heterogeneous catalyst [32,33] and/or proton conductors [34].

In previous reports we presented and explored the phosphonic acid [1,1′-biphenyl]-3,3′,5,5′-tetrakis(phosphonic acid) (H_8_btp) as a new tetraphosphonic organic ligand (Scheme 1) [28,31,34]. H_8_btp afforded a particular series of MOF-type compounds self-assembled from La^3+^ cations (or mixtures of lanthanides), which then showed not only fascinating structural features, but interesting photoluminescent properties. We also registered, however, that these 2D materials presented a significant drawback (which might also be seen as a disadvantage regarding a future practical application): their thermal stability was very low with the framework being only stable between ambient temperature and ca. 70 °C. In order to try to overcome this disadvantage, we have further explored the chelating ability of H_8_btp under different synthetic conditions, including a careful selection of the lanthanide radii, and isolated two new isotypical families of LnOFs which we report here:(i)layered [Ln_7_(H_5_btp)_4_(H_5.5_btp)_2_(H_6_btp)_2_(H_2_O)_12_]∙23.5H_2_O∙MeOH [where Ln^3+^ = Eu^3+^ (**1Eu**) and Gd^3+^ (**1Gd**)] (prepared using the same self-assembly methodology previously reported for the materials based on La^3+^ and H_8_btp, i.e., microwave-irradiation followed by slow evaporation);(ii)3D [Ln_4_(H_3_btp)(H_4_btp)(H_5_btp)(H_2_O)_8_]∙3H_2_O [where Ln^3+^ = Ce^3+^ (**2Ce**), Pr^3+^ (**2Pr**), and Nd^3+^ (**2Nd**)], obtained from conventional hydro(solvo)thermal synthesis. This approach allowed us to obtain a material with increased dimensionality (3D) in a much less synthesis time.

## 2. Results and Discussion

### 2.1. Crystal Structure Details

#### 2.1.1. [Ln_7_(H_5_btp)_4_(H_5.5_btp)_2_(H_6_btp)_2_(H_2_O)_12_]∙23.5H_2_O∙MeOH (1)

The structure of **1** was unveiled while studying single-crystals of [Eu_7_(H_5_btp)_4_(H_5.5_btp)_2_(H_6_btp)_2_(H_2_O)_12_]∙23.5H_2_O∙MeOH (**1Eu**) using X-ray diffraction. The isotypical nature of **1Eu** and **1Gd** was unequivocally confirmed by powder X-ray diffraction data (Appendix A), being supported by EDS mapping (Appendix A). The family crystallizes in the monoclinic space group P2_1_/*c*, with a large asymmetric unit composed of three and half metal centers and four different H_8−*x*_btp*^x^*^−^ residues (*x* = 2, 2.5 and 3). Structural adhesion is overall achieved by a complex network of hydrogen bonding interactions between the phosphonate groups and the several solvent molecules.

The four metal centers have different coordination numbers as depicted in Figure 1. Eu1 is at a center of inversion, with multiplicity of 2, being hexacoordinated to six phosphonate groups, while Eu2 is heptacoordinated to five different phosphonate groups and two coordination water molecules (O2W and O3W are disordered). The remaining two metal centers, Eu3 and Eu4 are also heptacoordinated to two coordination water molecules and five phosphonate groups, each. For the former two coordination environments, the phosphonate groups coordinate with the metal centers with the same k^1^-*O* mode of coordination while for the last two, besides this mode of coordination, two phosphonate groups have a k^2^-*O* coordination mode. The Eu–O bond lengths were found in the 2.221(7)–2.535(16) Å range, comparable to those reported for other Ln^3+^-based phosphonate compounds, and the internal O–La–O polyhedral angles are in the 69.5(4)–180.0(2)° range (Appendix A).

The phosphonic acid groups have three different protonation degrees: −3, −2.5 or −2. The partial protonation arises from the need to balance the overall crystal charge, and the location of the respective hydrogen atom was carefully selected and placed on O45 (based on proximity of hydrogen bonding acceptor. We further note that the structure also has hydrogen atoms which “jump” between adjacent phosphonate groups, namely between O23 and O30 and between O33 and O38.

As observed for the previously reported [Ln_4_(H_6_btp)_2_(H_4_btp)_2_(H_8_btp)(H_2_O)_16_]∙12H_2_O isotypical family of networks [where Ln^3+^ = La^3+^, (La_0.9_Eu_0.1_)^3+^ and (La_0.9_Tb_0.1_)^3+^] [28], **1** contains discrete 1D organic cylinders stacked in a typical brick-wall-like fashion in the *bc* plane, with solvent molecules occupying the available space in-between these cylinders as depicted in Figure 2. Connections between the organic cylinders are ensured by metallic dimers (*inset* in Figure 2a), leading to 2D layers placed in the *ac* plane of the unit cell, stacked on top of each other and maintained by weak π–π interactions between the aromatic rings of adjacent layers [distance of 3.982(6)Å and dihedral angle of 13.3(5)°].

Within these cylinders the organic linkers are stacked on top of each other with an overall rotation angle of ca. 135° along the [100] direction, forming a double-helix-like structural motif as depicted in Figure 3. Besides the connection to the metal centers, each organic residue forms weak π–π interactions between each other, with intercentroid distances ranging from 3.5805(4) to 3.8321(5) Å [dihedral angles between 4.8(4) and 6.1(5)°]. 

The presence of several phosphonate groups and coordinated and crystallization solvent molecules promote the formation of a complex network of hydrogen-bonding interactions. Within the organic cylinders, such interactions are of the P‒O⋯H‒O‒P type (i.e., inter-phosphonate groups) being rather strong and directional [*d_D⋯A_* distances between 2.496(13) and 3.263(13)Å with <(DHA) between 125–176°]. These ensure a close packing of the 2D layers and overall structural integrity of **1** (for more details see Appendix A).

It is interesting to note that the four independent organic linkers composing the aforementioned organic cylinder connect, each, to a different number of metal centers (3, 4, 5 and 6), ultimately leading to the 2D compact network which can be more easily analyzed from a pure topological perspective. Appendix A depicts the reduction of the network to simple nodes (i.e., the metal centers) and bridges (ensured by the organic linkers). Following the recommendations of Alexandrov at al. [35], who suggested that any moiety (ligand or atoms) connecting more than two metallic centers should be considered as a network node. For this case, both the metal centers and the organic linkers were considered as network nodes. We note that the hydrogen bonds were not included in these calculations. The network of the isotypical family of **1** is, therefore, an octanodal 3,4,5,5,5,5,6,6-connected network with total point symbol {4^13^·6^2^}_2_{4^3^}_2_{4^5^·6^3^·8^2^}_6_{4^6^·8^8^·10}{4^6^}_2_{4^9^·6}_2_ as revealed by the software package TOPOS [36]. Database review of the *Reticular Chemistry Structure Resource* (RCSR) [37] and in EPINET [38] show that this complex topology has not been described yet. We note that the network was attempted to be reduced to more simple nodes such as those described in the paper by Bonneau at al. [39] However, this approach did not significantly reduce the number of nodes ultimately leading instead to a mixture of different node approaches, hence in this manuscript we decided to maintain the classical approach. For this case, the notations {4^3^}, {4^9^·6}, {4^6^} correspond to the three H_8−*x*_btp*^x^*^−^ residues and {4^13^·6^2^}, {4^5^·6^3^·8^2^}, {4^6^·8^8^·10} corresponds to the metal centers. 

#### 2.1.2. [Ln_4_(H_3_btp)(H_4_btp)(H_5_btp)(H_2_O)_8_]∙3H_2_O (2)

The structure of the isotypical family of compounds [Ln_4_(H_3_btp)(H_4_btp)(H_5_btp)(H_2_O)_8_]∙3H_2_O [where Ln^3+^ = Ce^3+^ (**2Ce**), Pr^3+^ (**2Pr**), and Nd^3+^ (**2Nd**)] was elucidated for the La^3+^-based material. Besides elemental analysis (see Experimental Section), the homogeneity of the bulk samples was studied using powder X-ray diffraction (Appendix A), FT-IR spectroscopy (Appendix A), and electron microscopy (Appendix A).

Compound **2** crystallizes in the monoclinic space group P2_1_/*c*, with the asymmetric unit being composed of four metal centers and three different H_8−*x*_btp*^x^*^−^ residues (x = 5, 4 and 3). The four metal centers have different coordination numbers (Figure 4): La1 and La2 are both octocoordinated to five phosphonate groups and three coordination water molecules; La3 is nonacoordinated to seven different phosphonate groups and two coordination water molecules; La4 is heptacoordinated to six different phosphonate groups and one coordination water molecule. As expected, and compared with the structure of **1**, the overall coordination numbers are higher, thus reflecting the larger ionic radius of the metal centers. The La–O bond lengths were found in the 2.346(5)–2.851(6) Å range while the internal O–La–O polyhedral angles are in the 53.61(15)–59.40(15)° range (see Appendix A).

When compared to the previously described layered material, **2** has a smaller number of organic linkers in the asymmetric unit: a total of three, showing distinct protonation degrees −5, −4 and −3. The overall structural connectivity of each linker is distinct: one acts as a hexadentate bridge, connecting six metal centers, while the remaining residues act as a hepta- and octadentate linkers, connecting seven and eight metal centers each. For the former case, the phosphonate groups coordinate to the metal centers by simple k^1^-*O* coordination modes, while in the latter two situations there are also k^2^-*O* and μ-*O*,*O* modes of coordination involved. It is noteworthy to mention that when compared to the structural features of the organic linkers in **1**, one can easily infer that the bridges can easily accommodate more metal centers in the structure by increasing the overall connectivity and having, in average, a higher level of deprotonation.

The network of **2** is also composed by organic cylinders distributed in typical brick-wall-like fashion in the *ab* plane and formed by stacked organic linkers along [001] direction of the unit cell (Figure 5). The organic linkers are stacked, as depicted in Figure 6, on top of each other with a rotation angle of ca. 92° along the [100] direction, forming a double-helix-like structural motif, just as observed in the other related materials. As so, besides the connection with the metallic centers (described below), the organic linkers are further engaged in weak π–π interactions with intercentroid distances ranging between 3.5545(5) and 3.9692(5) Å [dihedral angles between 4 and 9°] (Figure 6). 

As for **1**, these cylinders are mutually connected by metallic dimers. However, unprecedented tetrameric units are also present for this family of compounds (*insets* in Figure 5a). Intermetallic distances: for the dimer, La2⋯La2 of 6.0643(6) Å; for the tetramer, La1⋯La3 of 4.0834(4) Å and 4.4179(5) Å. These metallic connections (covalent bonds), allied to very strong and directional hydrogen-bonding interactions between phosphonate groups [*d_D⋯A_* distances between 2.521(7) and 2.842(8)Å with <(DHA) between 111 and 172°—see Appendix A], are the main reason for the formation of the three-dimensional network of the MOF structure **2** as shown in Figure 5b.

Overall, this family of compounds has a lower number of both coordinated and crystallization solvent molecules. We note that these are also engaged in strong hydrogen bonding interactions with the phosphonate groups lining the inner walls of the channels of the network [*d_D⋯A_* distances between 2.553(9) and 3.013(8)Å with <(DHA) between 131 and 161°—see Appendix A].

Using the same topological strategy previously described for **1** (again, the hydrogen bonds were not included in the calculation), this new MOF structure is a new hexanodal 5,6,6,7,7,8-connected network with total point symbol {3·4^12^·6^2^}{3^2^·4^10^·6^9^}{3^4^·4^16^·5^6^·6^2^}{3^7^·4^11^·5^3^}_2_{4^4^·5·6·8^9^}{4^8^·6^2^} (see Appendix A), wherein {3·4^12^·6^2^},{ 3^2^·4^10^·6^9^},{ 3^4^·4^16^·5^6^·6^2^} and {3^7^·4^11^·5^3^}_,_ {4^4^·5·6·8^9^}, {4^8^·6^2^} notations correspond to the metal centers and H_8−*x*_btp*^x^*^−^ residues, respectively. We note that in this framework La1 and La3 exhibit the same point and extended symbol being, thus, equivalent hence the six-connectivity nature of the network.

### 2.2. Structural Characterization

#### 2.2.1. Thermogravimetry

The thermal stability of [Gd_7_(H_5_btp)_4_(H_5.5_btp)_2_(H_6_btp)_2_(H_2_O)_12_]∙23.5H_2_O∙MeOH (**1Gd**) was investigated between ambient temperature and ca. 800 °C, providing additional information on the hydration level of this family of isotypical compounds. As depicted in Appendix A, the thermogram for compound **1Gd** shares many similarities with that of [La_4_(H_6_btp)_2_(H_4_btp)_2_(H_8_btp)(H_2_O)_16_]∙12H_2_O [28], ultimately indicating a comparable thermal behavior. For **1Gd** the thermogram does not evidence unequivocal regions in which the mass remains stable. There is a continuous weight loss (more significant at some specific temperatures) over the entire temperature range. This behavior creates additional difficulties for individual assignments of temperature range *vs.* released residues, although it was foreseeable that this should happen due to the large number of coordination and crystallization solvent molecules present in the structure. One can, nevertheless, confirm the crystallographic studies for this compound. It is possible to discern three main weight losses: the first, between ambient temperature and ca. 100 °C, corresponding to 8.2% of the total weight loss, is attributed to the release of all crystallization solvent molecules from the structure (calculated 8.2%), i.e., the 23.5 water molecules plus the one methanol molecule. Above this temperature, there is an immediate second weight loss until ca. 200 °C. In this temperature range, the coordination water molecules are removed from the structure corresponding to a total weight loss of 4.3% (calculated 3.9%). The final step corresponds to the decomposition of the organic component.

The thermal behavior of the bulk [Ln_4_(H_3_btp)(H_4_btp)(H_5_btp)(H_2_O)_8_]∙3H_2_O (**2**) materials were also investigated between ambient temperature and ca. 800 °C. Due to the isotypical nature of the compounds the following paragraph discussion will be solely focused on compound **2Ce** (see Appendix A). For **2Ce**, one discerns three main weight losses. The first, between ambient temperature and ca. 110 °C, corresponding to 2.4% of the total weight loss, is attributed to the release of all crystallization solvent molecules (calculated 2.4%), i.e., a total of 3 water molecules. We emphasize that this value agrees well with the water content found from the performed crystallographic studies. Above this temperature, there is a second weight loss until ca. 250 °C. In this temperature range, the coordination water molecules are removed from the structure (in this case the observed 6.6% weight loss is equivalent to eight water molecules). The final step corresponds to the decomposition of the organic content.

Variable-temperature powder X-ray diffraction studies on [Ce_4_(H_3_btp)(H_4_btp)(H_5_btp)(H_2_O)_8_]∙3H_2_O (**2Ce**) further corroborate the aforementioned assumptions concerning the release of water molecules and phase modifications (Appendix A): from ambient temperature to ca. 300 °C the structure changes, leading to the formation of a new crystalline phase as a consequence of the release of crystallization and coordination water molecules. The crystalline structure starts to collapse above this temperature becoming almost fully amorphous at 550 °C.

While these compounds share many structural similarities, the MOF compound **2** shows superior framework stability when compared to the previously reported [Ln_4_(H_6_btp)_2_(H_4_btp)_2_(H_8_btp)(H_2_O)_16_]∙12H_2_O [28] and the herein reported [Ln_7_(H_5_btp)_5_(H_6_btp)_3_(H_2_O)_12_]∙23.5H_2_O∙MeOH (**1**) materials that became both amorphous at rather low temperatures. We attribute this feature to the increased connectivity between metal ions and organic linkers: in **2** the structure contains less solvent molecules, both coordinated and of crystallization, and a higher connectivity of the phosphonate groups; thus, the liberation of the solvent molecules, including those which may be coordinated to the metal centers and pointing towards the small channels of the network, do not promote such an earlier onset decomposition of the material, with crystalline material being observed at temperature as high as 450 °C (see *inset* in Appendix A).

#### 2.2.2. FT-IR Spectroscopic Studies

The vibrational FT-IR spectroscopy studies support the structural features unveiled by the X-ray diffraction studies. Appendix A show the FT-IR spectra of the materials in the 3650–350 cm^−1^ spectral region, including assignments for each of the main observed bands. Because **1** and **2** have the same functional groups, the spectra contain the same vibrational modes. A broad band is centered in the ca. 3600–2550 cm^−1^ spectral range, being attributed to both *ν*(O–H) stretching vibrational modes from coordination and crystallization water molecules and to the *ν*(PO–H) stretching vibration of the phosphonate groups. In the central region of the spectrum, between ca. 1745 and 1560 cm^−1^, it is possible to observe the typical *ν*(C=C) stretching vibrational modes arising from the aromatic rings and the deformation stretching of the water molecules. We further observe a sharp vibrational mode at ca. 695 cm^−1^, corresponding to the *ν*(P–C) stretching vibration. Also, in this region, the stretching vibrational modes of *ν*(P=O) were noticed from ca. 1300–1060 cm^−1^, and those of *ν*(P–O) from ca. 1070 to 830 cm^−1^.

### 2.3. Photoluminescence Studies

The excitation spectra of **1Eu** were recorded at ambient temperature (ca. 300 K) and 12 K monitoring the strongest Eu^3+ 5^D_0_→^7^F_2_ emission transition (Figure 7). Both spectra are dominated by a broad UV band (240–320 nm) attributed to π–π* transitions of the organic ligands, which are very similar to the one previous reported for the Ln^3+^ MOFs obtained with the H_8_btp organic linker [28,29,30,31]. The additional sharp lines in the spectra of **1Eu** are ascribed to the intra-4f^6 7^F_0,1_→^5^D_1-4_, ^5^L_6_, ^5^G_2-6_ and ^5^H_3-7_ transitions of Eu^3+^. The strong improvement of the 240–320 nm broad band, when the temperature decreases, demonstrates a non-effective energy transfer from the ligand to the Eu^3+^ ions (*antenna effect*) at the ambient temperature.

The emission spectra of **1Eu** recorded at ambient temperature and 12 K, excited at 393 nm are presented in Figure 8. The sharp Eu^3+^ emission lines are assigned to the ^5^D_0_→^7^F_0-4_ transitions. The ^5^D_1_→^7^F_0-3_ transitions are only properly noticed, even at 12 K, after magnification (Appendix A), demonstrating an efficient relaxation between the ^5^D_1_ and ^5^D_0_ excited states promoted by the vibrations of the organic linkers and the coordinated water molecules. The dominance of the ^5^D_0_→^7^F_2_ over the ^5^D_0_→^7^F_1_ transition is typical of Eu^3+^ environments without inversion center. Although, the structure of **1Eu** presents four independent Eu^3+^ crystal sites, one of which in an inversion center, the corresponding emission spectra do not allow discriminating the corresponding individual photoluminescence signatures. Nevertheless, changing the excitation wavelength from 393 to 310 nm at 12 K (inset of Figure 8) results on a substantial change on the emission profile, particularly on the ^5^D_0_→^7^F_1_ transition region. The emission spectrum obtained with the 310 nm excitation, in addition to the three lines of the 393 nm spectrum shows four additional Stark components, which is in good agreement with the presence of multiple Eu^3+^ local sites. In accordance, the ^5^D_0_ decay curve of **1Eu** under direct excitation at 393 nm (^5^L_6_ excited level), while monitoring the strongest ^7^F_2_ Stark component at ambient temperature (Appendix A), cannot be properly described by a single-exponential function. The data is only well fitted by a second order exponential function yielding lifetimes of 0.39±0.01 and 1.16±0.10ms, with an average lifetime of 0.52 ms (details in the Appendix A).

## 3. Materials and Methods

### 3.1. Synthesis of [Ln_7_(H_5_btp)_4_(H_5.5_btp)_2_(H_6_btp)_2_(H_2_O)_12_]∙23.5H_2_O∙MeOH (1)

Reactive mixtures composed of the respective lanthanide(III) chloride hydrates [LnCl_3_∙6H_2_O, where Ln^3+^ = Eu^3+^ (**1Eu**) and Gd^3+^ (**1Gd**)] and 0.025 g of [1,1′-biphenyl]-3,3′,5,5′-tetrayltetrakis(phosphonic acid) (**H_8_btp**), with an overall molar ratio of approximately 1:4 (H_8_btp: Ln^3+^), were individually prepared at ambient temperature in a mixture of distilled water, HCl (6 M) and methanol (2 mL each solvent), and placed inside a 10 mL IntelliVent microwave reactor. The reaction took place inside a CEM Focused Microwave Synthesis System Discover S-Class equipment, under constant magnetic stirring (controlled by the microwave equipment), at 100 °C for 60 min (using an irradiation power of 50 W). A constant flow of air (ca. 20–30 psi of pressure) ensured a close control of the temperature inside the reactor. After irradiation, the reactors with the homogeneous solutions were uncapped and left motionless until ca. 3/4 of the reaction mixture volume had evaporated. This methodology successfully led to the preparation of [Ln_7_(H_5_btp)_4_(H_5.5_btp)_2_(H_6_btp)_2_(H_2_O)_12_]∙23.5H_2_O∙MeOH (**1Eu** and **1Gd**) materials as white crystalline plates. The resulting compounds were recovered by vacuum filtration, washed with copious amounts of distilled water and then air-dried at ambient temperature.


**For [Gd_7_(H_5_btp)_4_(H_5.5_btp)_2_(H_6_btp)_2_(H_2_O)_12_]∙23.5H_2_O∙MeOH**


Elemental CH composition (%): Calcd: C 21.0; H 3.01. Found: C 19.9; H 3.02.

Thermogravimetric analysis (TGA) data (weight losses in %) and derivative thermogravimetric peaks (DGT; in italics inside parentheses): 25–100 °C (89 °C) −8.2%; 100–200 °C (179 °C) −4.3%; 200–440 °C (410 °C) −7.4%; 440–800 °C (503 °C) −20.1% 

Selected FT-IR data (in cm^−1^; from KBr pellets): ν(H_2_O + POH) = 3600–2800br; ν(C=C) + δ(H_2_O) = 1745–1560m; ν(P=O) = 1300–1070vs; ν(P–O) = 1070–830vs; ν(P–C) = 695m.

### 3.2. Synthesis of [Ln_4_(H_3_btp)(H_4_btp)(H_5_btp)(H_2_O)_8_]∙3H_2_O (2)

Reactive mixtures composed of the respective lanthanide(III) chloride hydrates [LnCl_3_∙6H_2_O, where Ln^3+^ = Ce^3+^ (**2Ce**), Pr^3+^ (**2Pr**), and Nd^3+^ (**2Nd**)] and 0.025 g of [1,1′-biphenyl]-3,3′,5,5′-tetrayltetrakis(phosphonic acid) (**H_8_btp**), with an overall molar ratio of approximately 1:4 (H_8_btp: Ln^3+^), were individually prepared in a mixture of distilled water, HCl (6 M) and methanol (2 mL each solvent). Mixtures were kept under constant magnetic stirring in open air and ambient temperature for approximately 15 min. The resulting homogeneous suspensions were transferred to Teflon-lined Parr Instrument reaction vessels and placed inside an MMM Venticell oven. The heating program included: (i)heating during 48 h up to 140 °C;(ii)24 h uphold of the temperature (140 °C);(iii)cooling down during 48 h until ambient temperature.

The resulting materials were isolated as white microcrystalline powders, recovered by vacuum filtration, washed with abundant amounts of distilled water and dried at ambient temperature. Single-crystals of this compound were obtained from different experimental conditions (denoted as **2La**—see the Appendix A for further details).


**Elemental CH composition (%):**


Calcd for **2Ce**: C 19.9; H 2.42. Found: C 21.0; H 2.47.

Calcd for **2Pr**: C 19.9; H 2.41. Found: C 20.7; H 2.37.

Calcd for **2Nd**: C 19.8; H 2.40. Found: C 20.7; H 2.41.

Thermogravimetric analysis (TGA) data (weight losses in %) and derivative thermogravimetric peaks (DTG, in italics inside the parentheses):

**2Ce**: 25–200 °C −18.05% (110 °C); 200–400 °C −5.98% (271 °C). Total loss: 24.0%.

**2Pr**: 25–122 °C –10.13% (109 °C); 122–200 °C –5.60% (135 °C); 200–294 °C –6.05% (270 °C); 294–400 °C –6.51% (324 °C). Total loss: 28.3%.

**2Nd**: 25–133 °C −12.78% (108 °C); 133–205 °C –5.92% (144 °C); 205–367 °C −6.58% (281 °C). Total loss: 25.3%.

Selected FT-IR data (in cm^−1^; from KBr pellets):

**2Ce**: ν(H_2_O + POH) = 3600–2550br; ν(C=C) + δ(H_2_O) = 1700–1570m; ν(P=O) = 1270–1060vs; ν(P–O) = 945–840vs; ν(P–C) = 696m.

**2Pr**: ν(H_2_O + POH) = 3610–2570br; ν(C=C) + δ(H_2_O) = 1715–1540m; ν(P=O) = 1245–1070vs; ν(P–O) = 945–850vs; ν(P–C) = 696m.

**2Nd**: *ν*(H_2_O + POH) = 3625–2565*br*; *ν*(C=C) + *δ*(H_2_O) = 1695–1560*m*; *ν*(P=O) = 1245–1070*vs*; *ν*(P–O) = 940–850*vs*; *ν*(P–C) = 696*m*.

### 3.3. Single-Crystal X-ray Diffraction Studies

Single crystals of compounds [Eu_7_(H_5_btp)_4_(H_5.5_btp)_2_(H_6_btp)_2_(H_2_O)_12_]∙23.5H_2_O∙MeOH (**1Eu**) and [La_4_(H_3_btp)(H_4_btp)(H_5_btp)(H_2_O)_8_]∙3H_2_O (**2La**) were manually harvested from the crystallization vials and immersed in highly viscous FOMBLIN Y perfluoropolyether vacuum oil (LVAC 140/13, Sigma-Aldrich) to avoid degradation caused by the evaporation of the solvent [40]. Crystals were mounted on either Hampton Research CryoLoops or MiTeGen MicroLoops, typically with the help of a Stemi 2000 stereomicroscope equipped with Carl Zeiss lenses.

X-ray diffraction data for **1Eu** and **2La** were collected at 150(2)K on a Bruker D8 QUEST equipped with Mo Kα sealed tube (λ = 0.71073 Å), a multilayer TRIUMPH X-ray mirror, a PHOTON 100 CMOS detector, and an Oxford Instruments Cryostrem 700+ Series low temperature device. Diffraction images were processed using the software package SAINT+ [41], and data were corrected for absorption by the multiscan semi-empirical method implemented in SADABS 2016/2 [42]. Structures were solved using the algorithm implemented in SHELXT-2014/5 [43], which allowed the immediate location of almost all of the heaviest atoms composing the molecular unit of the two compounds. The remaining missing and misplaced non-hydrogen atoms were located from difference Fourier maps calculated from successive full-matrix least-squares refinement cycles on *F*^2^ using the latest SHELXL from the 2018/3 release [44]. All structural refinements were performed using the graphical interface ShelXle [45].

For **1Eu**, the hydrogen atoms bound to carbon were placed at their idealized positions using the *HFIX 43* in SHELXL-2014, which were included in subsequent refinement cycles with isotropic displacement parameters (*U*_iso_) fixed at 1.2×*U*_eq_ of the parent carbon atoms. Hydrogen atoms associated with the terminal –POH groups were placed at their idealized positions using the *HFIX 83* instruction and were refined assuming isotropic thermal displacements parameters (*U*_iso_) fixed at 1.5×*U*_eq_ of the parent oxygen atoms. To find which groups should be protonated it was necessary to simultaneously take into account the P–O bond lengths and the possibility of forming hydrogen bonds with neighboring moieties. Two of these –POH groups shared the hydrogen atom with the neighboring phosphonic group. O23 and O33 shared the hydrogen atom with O30 and O38, respectively. These hydrogen atoms were refined with a 50% occupancy rate each.

The crystallization methanol molecule was refined with 50% of occupancy rate. The hydrogen atoms from the methyl and hydroxyl groups were placed at their idealized positions using the *HFIX 137* and *HFIX 83*, respectively, and were included in subsequent refinement cycles with isotropic displacement parameters (*U*_iso_) fixed at 1.5×*U*_eq_ of the parent non-hydrogen atoms.

Due to the considerable disorder of the crystallization water molecules, their hydrogen atoms could not be located from difference Fourier maps and no attempts were made to place them in calculated positions. These atoms were, however, included in the empirical formula of the compound (Table 1). The disorder of these water molecules was refined as follow: 30% occupancy rate for O16W and O19W; 40% for O15W, O21W, O22W and O23W; 50% for O10W, O20W, O25W and O26W; 60% for O14W and O17W; 70% for O13W; 80% for O24W; and 85% for O12W. These occupancy factors were fixed in the final structural model but were obtained from a previous unrestrained refinement for each site. All oxygen atoms belonging to the water molecules of crystallization were refined using a common isotropic displacement parameter.

For **2La** the hydrogen atoms bound to carbon were placed at their idealized positions using the *HFIX 43* in SHELXL-2014, which were included in subsequent refinement cycles with isotropic displacement parameters (*U*_iso_) fixed at 1.2×*U*_eq_ of the parent carbon atoms. Hydrogen atoms associated with the terminal –POH groups were placed at their idealized positions using the *HFIX 147* instruction and were refined assuming isotropic thermal displacements parameters (*U*_iso_) fixed at 1.5×*U*_eq_ of the parent atoms. Two crystallization water molecules, namely O11W and O12W, were disordered and refined with an occupancy rate of 20% and 80% respectively. The hydrogen atoms of all water molecules could not be located from difference Fourier maps and no attempts were made to place them in calculated positions. These atoms were, however, included in the empirical formula of the compound.

The last difference Fourier map synthesis showed: for **1Eu** the highest peak (2.005 eÅ^−3^) located at 0.65 Å from Eu1 and the deepest hole (−2.108 eÅ^−3^) located at 0.13 Å from O24W, while compound **2La** showed the highest peak (2.468 eÅ^−3^) and the deepest hole (−1.888 eÅ^−3^) located at 0.56 Å and 0.90 Å from La4, respectively. Structural drawings have been created using the software package Crystal Impact Diamond [46].

Information concerning crystallographic data collection and structure refinement details is summarized in Table 1. Appendix A gather the most significant distances and angles of the lanthanide coordination environments and hydrogen bonding geometric details. Crystallographic data (including structure factors) for the crystal structures of **1Eu** and **2La** have been deposited with the Cambridge Crystallographic Data Centre as supplementary publication data No. 1,964,385 and 1964384, respectively. Copies of the data can be obtained free of charge on application to CCDC, 12 Union Road, Cambridge CB2 2EZ, U.K. FAX: (+44) 1223 336033. E-mail: deposit@ccdc.cam.ac.uk.

## 4. Conclusions

The rigid tetrapodal organic linker [1,1′-biphenyl]-3,3′,5,5′-tetrayltetrakis(phosphonic acid) (**H_8_btp**) was used for the preparation of two new isotypical families of lanthanide–organic frameworks: Ln_7_(H_5_btp)_4_(H_5.5_btp)_2_(H_6_btp)_2_(H_2_O)_12_]∙23.5H_2_O∙MeOH [where Ln^3+^ = Eu^3+^ (**1Eu**) and Gd^3+^ (**1Gd**)], and [Ln_4_(H_3_btp)(H_4_btp)(H_5_btp)(H_2_O)_8_]∙3H_2_O [where Ln^3+^ = Ce^3+^ (**2Ce**), Pr^3+^ (**2Pr**), and Nd^3+^ (**2Nd**)]. This work complements our past research efforts using this ligand, particularly that which described in detail the photoluminescent family [Ln_4_(H_6_btp)_2_(H_4_btp)_2_(H_8_btp)(H_2_O)_16_]∙12H_2_O [where Ln^3+^ = La^3+^, (La_0.9_Eu_0.1_)^3+^ and (La_0.9_Tb_0.1_)^3+^] [28].

In our previous report we employed microwave-assisted synthesis at 100 °C (60 min at 50 W) followed by slow evaporation over a period of 7 days. Family **1** herein reported followed an identical synthetic procedure, while trying to explore the structural effect when using smaller radii for the lanthanide cations: we note that in our past work photoluminescent materials were only obtained by doping an inert La^3+^-based matrix. Two-dimensional compact layers were again obtained, in which the ligand-to-metal ratio is greater than unity. This is an important structural aspect because it means that, despite both synthesis contain in the reactive mixtures a great excess of metal centers, the microwave heating procedure followed by slow evaporation seems to favor the inclusion of a large number of solvent molecules, increasing concomitantly the number of supramolecular contacts in the structures. To note that the same experimental conditions using the hydrothermal method do not permit the preparation of this material. As described for [Ln_4_(H_6_btp)_2_(H_4_btp)_2_(H_8_btp)(H_2_O)_16_]∙12H_2_O, compound **1** is also formed by compact layers in which organic cylinders are interconnected by the metallic centers. The presence of optically active Eu^3+^ lanthanide centers yielded photoluminescent materials. As for our previous reported material, the presence of several crystallization and coordinated water molecules in the structure accounts for the lower emission of the Eu^3+^-based material.

Family **2** of compounds is formed by a three-dimensional network in which, comparatively, the ligand-to-metal ratio is below the unity. We achieved this by increasing the temperature to 140 °C and change the synthetic method to the traditional hydro(solvo)thermal one. This strategy, allied to the usage of larger lanthanides, seemed to favor metallic connectivity in detriment to the various supramolecular contacts involving solvent molecules. This new 3D family of compounds is significantly more compact, but it is still formed by organic cylinders interconnected by the metallic centers, as for family **1** and our previously reported [Ln_4_(H_6_btp)_2_(H_4_btp)_2_(H_8_btp)(H_2_O)_16_]∙12H_2_O compounds.

These two families of compound clearly show the importance of a careful selection of the synthetic experimental method. Despite both syntheses start with the same reactive mixtures (with a great excess of metal), the microwave heating procedure followed by slow evaporation seems to favor the inclusion of a large number of solvent molecules. This leads to an increasing of the number of supramolecular contacts in the structures, originating a 2D layered material. On the other hand, hydrothermal synthesis at higher temperature seems to favor metal coordination of a higher number of phosphonic groups, leading to a more compact 3D network. 

Future work related to this system is now being currently focused at the selective control of the dimensionality of the networks so to explore various properties, particularly those related with conductivity: the presence of solvent molecules and many protonated phosphonate groups is expected to significantly boost proton conductivity.

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
