# Peer review of "Coordination Polymers Based on a Biphenyl Tetraphosphonate Linker: Synthesis Control and Photoluminescence"

_molecules, 2020, doi:10.3390/molecules25081835_

Round 1
Reviewer 1 Report
In their contribution, Almeida Paz et al. describe the syntheses and structural characterizations of a series of lanthanide based MOFs which are derived from a tetratopic phosphonic acid ligand. The new material are characterized by Xray diffraction, TGA and IR Emission properties are finally provided and discussed.
The use of phosphonate ligands for the synthesis of MOF is appealing because they tend to form strong metal-ligand bonds compared to the more conventional carboxylate ligands counterparts. However their synthesis is also more challenging, often leading to dense or amorphous materials. In that regard the work present here is interesting, and the author provide a thorough structural characterization of the obtained La-MOFs with solid in depth discussion of their structural features.
The originality of the paper is however lowered by the fact the the work is mainly an extension of already published studies (from the same authors) describing MOFs obtained from the same phosphonic acid ligand but with other lanthanide metals centers (Ref 22; Inorg. Chim. Acta 2017, 455, 584).
It would have been interesting for example to investigate the chemical stability of these frameworks in acidic/basic aqueous conditions.
Author Response
We thank the reviewer for the positive comments.
The sole question raised at the end is focused at the stability of the materials in acid/basic medium. These were not tested for these particular compounds because of the properties which were investigated. However, it is our vast experience in lanthanide phosphonates that the materials tend to be slightly resilient to mild acid/basic conditions but typically are destroyed at more harsh conditions.
Reviewer 2 Report
This article successfully synthesized a series of lanthanide metal organic complexes based on rigid tetrapodal organic ligands by microwave method, normal temperature volatilization method and hydrothermal method. In addition, the structure and properties of these compounds have also been characterized. I failed to see crystallographic data and checkcif report due to damage to the compressed package of supplementary materials. Please forgive me for failing to make a reasonable judgment of the crystal structure data and molecular formula. This article is not bad, but there are still some problems. I have listed them below.
- In the introduction, it is recommended to supplement the relevant content of rare earth ion radius and ligand matching, and attach literature.
- There are many factors that affect the thermal stability of the complex. It is recommended that the relevant content be reasonably modified.
- It is recommended to add the angle between the planes with pi-pi interaction.
- Please unify the number of significant digits and the number of uncertain digits.
- In section 2.1, it is recommended to briefly explain the nodes of the topology.
- In section 2.3, please explain or calculate the description of LMCT. Please explain why 5D1-7F0 does not appear in the emission spectrum.
- In section 3.3 and Table 1 Please modify La to the correct name.
- It is recommended to supplement the excitation and emission spectra of compound 2 series. Because compound series 2 has fewer water molecules and solvent molecules, the “the presence of several crystallization and coordinated water molecules in the structure accounts for the lower emission of the Eu3+-based material” mentioned in the conclusion can be corroborated from the compound 2 optical data.
- Please explain why the error in the peaks of the two curves in Figure S1 (b).
Author Response
Q: In the introduction, it is recommended to supplement the relevant content of rare earth ion radius and ligand matching, and attach literature.
A: A section regarding the rare-earth ionic radius was included in the revised manuscript.
Q: It is recommended to add the angle between the planes with pi-pi interaction.
A: As requested by the reviewer, the dihedral angles between average planes have been added throughout the manuscript.
Q: Please unify the number of significant digits and the number of uncertain digits.
A: The number of significant and uncertain digits are represented in accordance with the values obtained by the crystallographic data. This depends significantly on the overall diffraction quality and the integration of the crystallographic data.
Q: In section 2.1, it is recommended to briefly explain the nodes of the topology.
A: The network nodes used for the studies were further explained and the point symbols of each was fragmented and assigned to its components in the revised manuscript.
Q: In section 2.3, please explain or calculate the description of LMCT. Please explain why 5D1-7F0does not appear in the emission spectrum.
A: Motivated by the comments from the reviewer we have revised the referenced literature on the study of the LMCT in Eu3+compounds. We concluded that it is unlikely the attribution of the observed band to a ligand-to-Eu3+charge transfer (CT) band. Moreover, the observed broad UV band is very similar to those reported by us for Tb3+-based MOFs containing the H8btp linker (refs. 22 and 25 of the manuscript). On the other hand, in accordance with for instance Dorenbos (J. Phys.: Condens. Matter, 2003, 15, 8417-8434) the CT bands for Tb3+cannot be observed in this energy range, they are expected to appear at considerable higher energies, on the range of ca. 8 eV (155 nm). In agreement with the reference the LMCT was suppressed in the manuscript.
The energy vibrations of the organic linker (such as from C-H, N-H and O-H oscillators) easily match the small energy difference between the 5D1and 5D0Eu3+excited states promoting an efficient relaxation process between these stats. Consequently, commonly the 5D1→7FJtransitions of Eu3+containing MOFs and complexes are not observed or are very weak relatively to the 5D0→7FJtransitions. For the MOF 1Euthe 5D1→7F0-3transitions can only be properly identified after magnification in the emission spectrum collected at 12 K (Figure below). This Figure was included in the SI as the new Figure S15 and discussed in the manuscript.
Q: In section 3.3 and Table 1 Please modify La to the correct name.
A: The term La is correct. We note that while the family is comprised of Ce3+, Pr3+and Nd3+, the crystal was obtained from a synthesis with La3+. However, the La3+compound is not associated to this family because the synthesis did not allow the isolation of compound 2as a pure phase.
Q: It is recommended to supplement the excitation and emission spectra of compound 2 series. Because compound series 2 has fewer water molecules and solvent molecules, the “the presence of several crystallization and coordinated water molecules in the structure accounts for the lower emission of the Eu3+-based material” mentioned in the conclusion can be corroborated from the compound 2 optical data.
A: Despite compounds of family 2 present a fewer number of water molecules and solvent molecules in the compound formulation, the average number of coordinated water molecules per each Ln3+is higher in family 2than in family 1(2 versus 1.7). In addition, contrary to the family 1which has one Ln3+without coordinated water molecules, all Ln3+sites of family 2have, at least, one coordinated water molecule. It is well known that the water molecules, through the O-H oscillators, promotes the non-radiative relaxation of excited states having a strong quenching effect on the Ln3+emission capabilities. The Ln3+ions with lower energy differences between the emitting and fundamental levels (near infrared emitters such as Nd3+) or with multiple emitting states with consecutive low energy separation such as in Pr3+are much more sensitive to the non-radiative relaxation effect of water molecules than for instance Eu3+, which presents a reasonable energy separation between their emitting and fundamental levels. Relevant Ce3+luminescence was only also reported for inorganic water free compounds. In the conclusions we state, “As for our previous reported materials, the presence of several crystallization and coordinated water molecules in the structure accounts for the lower emission of the Eu3+-based material”,indicating that MOF 1Eu presents lower emission capabilities than the parent materials of refs. 22 and 25 – the relative lower differences of the intensities between the intra 4f lines and the ligands UV broad band reveals that. Consequently, as tested at room temperature, it is not expected to observe photoluminescence for the compounds of series 2 with Nd3+, Pr3+and Ce3+.
Q: Please explain why the error in the peaks of the two curves in Figure S1 (b).
A: We thank the reviewer for this comment which clearly may raise some questions. The data collected and presented was repeated several times being always coherent. We note that single-crystal data was collected directly from crystals isolated in the mother liquor. Collecting powder X-ray diffraction data requires drying slightly the crystals and grind them to a powder. These two processes were repeated in a very gentle fashion but they seem to promote, in our opinion, a small modification of the overall structural of the network. Attempts to collect single-crystal data from the same dried crystals typically resulted in poor diffraction but with identical unit cell parameters. Thus, we believe that the differences in intensities arise from a re-organisation of the solvent molecules within the unit cell which, concomitantly, may slightly modify as well the structure of the coordination polymer. As mentioned, this is not a sporadic effect but a repeatable one, hence the differences registered.
Reviewer 3 Report
See attached.

Author Response
Q: Page 1, Line 37:
The authors refer to designing networks with “specific sizes and shapes.” This language is a little bit misleading – the network itself doesn’t have “shape” or “size;” rather, features in the PCP (e.g., pores) have defined size and shape. The authors should clarify their language here.
A: As the reviewer commented, this was to be referred as a feature of PCP, in which the pores can be designed with different shapes and sizes. The misleading language has been modified accordingly.
Q: Page 1, Line 43:
Instead of writing out “Metal-Organic Frameworks,” the authors should use the abbreviation “MOFs,” as this abbreviation is already defined earlier in the manuscript.
A: The abbreviation was included in this line. We thank the reviewer for pointing out this typo.
Reviewer 4 Report
The paper is an interesting (yet not novel though) contribution to the literature on emerging class of tetraphosphonates, which forms the basis for construction of functional metal-organic frameworks and coordination polymers based. Metal-phosphonates represent early examples of organic-inorganic hybrid materials, as well as materials with an extensive structural chemistry and a wide range of useful physical properties.
Authors clearly present two new families of compounds and discuss various structural properties. They also tried to report new topology of the complicated coordination net (comment below). Among relevant properties associated with lanthanide centres themselves, they investigated steady-state and time-resolved properties of the europium analogue.
I think that this contribution largely deserves publication. Nevertheless I think that three aspects of this work should be improved before publication:
1.Topology
One of CPs obtained by Authors has the formula [Ln4(H6btp)2(H4btp)2(H8btp)(H2O)16]∙12H2O, while the topological notation (NOT topology) of their octanodal net goes as follows: {413·62}2{43}2{45·63·82}6{46·88·10}{46}2{49·6}2. Authors should assign point symbol of a given node to a given fragment of the structure, since the net point symbol withut clarification of each of its components simply tells nothing. For each node the connectivity should be clearly assigned. See https://pubs.acs.org/doi/pdf/10.1021/cg501348g as example how the "decomposition" of topological notation can be conducted for a rather complicated net.
The same must be done for [Ln4(H3btp)(H4btp)(H5btp)(H2O)8]∙3H2O family of compounds.
Moreover, authors did not clearly indicate whether they included hydrogen bonds as topological paths. Please state clearly.
2.Slightly misleading (or incomplete) title
The term "tetraphosphonate" in the phosphonate community, by a rule of thumb, is associated with tetraphosphonates based on tetraphenylmethane and related cores (see next section for details). Thus, the title from the very beginning elicits false hopes pertaining to its content, so I strongly suggest adding "biphenyl" to the title, just to be clear.
3. Broader introduction into tetraphosphonates
Authors use "tetraphosphonates" in the title clearly due to the rich content of phosphonate groups in their compound.
Nevertheless, is should be stressed that at present there is already large amount of literature, dating back at least since 2013, on tetraphosphonates based on tetraphenylmethane (and later, tetraphenylsilane). Quite surprisingly, the introduction provides no broader context, not a single reference on tetraphosphonate ligands of this type, although this field has been comprehensively reviewed at least two times. Thus, I request updating the introduction section with some key literature of tetraphosphonates based on tetraphenylmethane, i.e. have a look a little beyond their own field.
Finally it should be stated why photoluminescent properties of compounds comprising Pr3+ and Nd3+ were not investigated.
Author Response
Q: One of CPs obtained by Authors has the formula [Ln4(H6btp)2(H4btp)2(H8btp)(H2O)16]∙12H2O, while the topological notation (NOT topology) of their octanodal net goes as follows: {413·62}2{43}2{45·63·82}6{46·88·10}{46}2{49·6}2. Authors should assign point symbol of a given node to a given fragment of the structure, since the net point symbol without clarification of each of its components simply tells nothing. For each node the connectivity should be clearly assigned. See https://pubs.acs.org/doi/pdf/10.1021/cg501348g as example how the "decomposition" of topological notation can be conducted for a rather complicated net. The same must be done for [Ln4(H3btp)(H4btp)(H5btp)(H2O)8]∙3H2O family of compounds.
A: The point symbols of each node was presenred and assigned to its components.
Q: Moreover, authors did not clearly indicate whether they included hydrogen bonds as topological paths. Please state clearly.
A: The hydrogen bonds were not taken into consideration for the topological studies. This was further clarified in the revised manuscript.
Q: Authors use "tetraphosphonates" in the title clearly due to the rich content of phosphonate groups in their compound. Nevertheless, is should be stressed that at present there is already large amount of literature, dating back at least since 2013, on tetraphosphonates based on tetraphenylmethane (and later, tetraphenylsilane). Quite surprisingly, the introduction provides no broader context, not a single reference on tetraphosphonate ligands of this type, although this field has been comprehensively reviewed at least two times. Thus, I request updating the introduction section with some key literature of tetraphosphonates based on tetraphenylmethane, i.e. have a look a little beyond their own field.
A: A section regarding tetraphosphonates was complemented in the introduction of the manuscript.
Q: Finally it should be stated why photoluminescent properties of compounds comprising Pr3+ and Nd3+ were not investigated
A: The compounds comprising Pr3+, Ce3+and Nd3+were tested at ambient temperature, but no photoluminescence properties was observed for the compounds.
Round 2
Reviewer 4 Report
Authors responded to all suggestions, and improved / clarified everything what was needed. I recommend "accept as is" decision.